

# Rényi entropies and negative central charges in non-Hermitian quantum systems

**Yi-Ting Tu[1,2]⋆, Yu-Chin Tzeng[1]† and Po-Yao Chang[1]‡**

**1** Department of Physics, National Tsing Hua University, Hsinchu 30013, Taiwan
**2** Department of Physics, University of Maryland, College Park, MD, USA

⋆ yttu@umd.edu , † yctzeng@mx.nthu.edu.tw , ‡ pychang@phys.nthu.edu.tw

## Abstract

Quantum entanglement is one essential element to characterize many-body quantum systems. However, the entanglement measures are mostly discussed in Hermitian systems. Here, we propose a natural extension of entanglement and Rényi entropies to non-Hermitian quantum systems. There have been other proposals for the computation of these quantities, which are distinct from what is proposed in the current paper. We demonstrate the proposed entanglement quantities which are referred to as *generic* entanglement and Rényi entropies. These quantities capture the desired entanglement properties in non-Hermitian critical systems, where the low-energy properties are governed by the non-unitary conformal field theories (CFTs). We find excellent agreement between the numerical extrapolation of the negative central charges from the *generic* entanglement/Rényi entropy and the non-unitary CFT prediction. Furthermore, we apply the *generic* entanglement/Rényi entropy to symmetry-protected topological phases with non-Hermitian perturbations. We find the *generic* $n$-th Rényi entropy captures the expected entanglement property, whereas the traditional Rényi entropy can exhibit unnatural singularities due to its improper definition.



# 1   Introduction

Boltzmann's entropy is the revolutionized formula that connects the observable of a macrostate to the probability distribution of possible microstates. This concept has been extended to quantum systems, where the von Neumann (entanglement) entropy measures the entanglement of a many-body quantum state. In modern condensed matter physics, these entanglement measures, including the entanglement entropy and the Rényi entropies provide a deep insight of diagnosing and characterizing quantum phases of matter. In particular, for critical systems in $(1+1)$ dimensions, the entanglement entropy has a universal scaling $S_A \sim \frac{c}{3} \ln L_A$, with $L_A$ being the length of subsystem $A$ and $c$ being the central charge of the corresponding conformal field theory (CFT) [1–4].

To date, the interest in entanglement has been mainly focused on Hermitian quantum systems. A systematic analysis of entanglement properties in non-Hermitian quantum systems [5–10] is still desired. In this article, we fill up this gap by proposing the *generic* entanglement and Rényi entropies which capture the desired entanglement properties in non-Hermitian quantum systems. To illustrate the validity of our proposal, we first emphasize one issue of entanglement measures in non-Hermitian systems. In critical non-Hermitian systems which are governed by non-unitary CFTs, the negative central charge can lead to the negative entanglement entropy [11–16]. The negative entanglement entropy seems problematic because the reduced density matrix is positive semi-definite which cannot give rise to a negative value of the entanglement entropy. To reconcile this issue, Refs. [17–19] suggest that the entanglement entropy is still positive and the true central charge is replaced by an *effective* central charge $c_{\text{eff}}$ which is positive[1] [17].

Alternatively, one can define a reduced density matrix which involves the left and right eigenstates in non-Hermitian systems. In this definition, the entanglement entropy is no longer guaranteed to be positive and leads to a possibility to obtain the negative central charge. By using this definition of the reduced density matrix combined with a modified trace, Ref. [20] has shown the negative central charge can be obtained in one-dimensional quantum group symmetric spin chains. In Ref. [21], the authors demonstrate that with the null vector condition on the twist fields in the cyclic orbifold, the Rényi entropy can be negative and the corresponding negative central charge can be obtained in the non-Hermitian conformal field theories[2]. In addition, Ref. [22] shows that the negative central charge can be obtained by

---

[1]In these approaches, the left and right eigenstates coincide at critical points due to PT symmetry together with chiral factorization of CFT. It leads to the usual definition of the reduced density matrix and gives the positive value of the entanglement entropy. The corresponding effective central charge is $c_{\text{eff}} = c - 24h$, where $h$ is the conformal weight of the ground state with $h < 0$, i. e., the ground state is not the conformal vacuum. If one set $h = 0$, the entanglement entropy can scale with the *true* central charge $c$

[2]It is interesting to point out that the Rényi entropy of the ground state with conformal weight $h < 0$ by

choosing proper branch cuts in the calculation of the entanglement entropy in the free-fermion models. In complementary to these existing approaches, we propose a natural extension of entanglement/Rényi entropy, which we referred to as the *generic* entanglement/Rényi entropy. The *generic* entanglement/Rényi entropy not only captures the desired negative central charge in several non-Hermitian critical systems, but also applicable to gapped symmetry protected topological phases. For the former critical cases, by using the logarithmic scaling property of the *generic* entanglement/Rényi entropy, we numerically obtain the negative central charge in the two-legged Su-Schrieffer-Heeger (SSH) model at the critical points and the q-deformed XXZ model with imaginary boundary terms. For the latter case, we demonstrate that the *generic* entanglement/Rényi entropy is a smooth function of the Hermitian breaking parameter in the Affleck-Kennedy-Lieb-Tasaki (AKLT) model, whereas the traditional Rényi entropy has unphysical singularities. Thus, the *generic* entanglement and Rényi entropies provide an unambiguous way of extracting the entanglement properties in non-Hermitian systems.

## 2  Generic Entanglement entropy and Rényi entropy

In non-Hermitian quantum systems, the density matrix $\rho = \sum_{\alpha\beta} \rho_{\alpha\beta} |\psi_\alpha^R\rangle\langle\psi_\beta^L|$ can be defined by the left and right biorthogonal states with $\langle\psi_\alpha^L|\psi_\beta^R\rangle = \delta_{\alpha\beta}$. Here, the density matrix inherits the non-Hermicity of the Hamiltonian $\rho^\dagger \neq \rho$. Suppose $\rho$ has nonnegative and real eigenvalues, the expectation value of an observable $O$ is defined as $\langle O\rangle = \text{Tr}(\rho O)$ [23]. The expectation value of $O$ is interpreted as the probabilistic expected value of the measure of $O$. For local observable $O_A$ in subsystem $A$, the density matrix is replaced by the reduced density matrix $\rho_A = \text{Tr}_{\bar{A}}\rho$ for measuring the expected outcomes. Here $\bar{A}$ denotes the complementary part of $A$. However, in non-Hermitian quantum systems, the eigenvalues of $\rho_A$ can be negative or even complex. This indicates the probability interpretation of the eigenvalues of $\rho_A$ must be extended to negative or complex numbers. In physics, we often require a measurable quantity to be a real number, which leads to certain constraints of the measurable quantity. The entanglement properties are the measures (the expectation values) of the "entanglement" in a quantum system. For example, the entanglement entropy is defined as the expectation value of the logarithm of the probability of states in the subsystem $A$, $S_A = -\text{Tr}(\rho_A \ln\rho_A) = \langle -\ln\rho_A\rangle$. One can generalize the entanglement measures to other quantities such as the $n$-th Rényi entropy $S_A^{(n)}$, $\exp((1-n)S_A^{(n)}) = \langle\rho_A^{n-1}\rangle$.

Since $\rho_A$ is generically non-Hermitian, the usual entanglement measures will not be real. One needs to define a more generic form of entanglement measures that are applicable for both Hermitian and non-Hermitian quantum systems. For a non-Hermitian reduced density matrix $\rho_A$, we decompose its eigenvalues $\omega_\nu$ into the amplitude and the phase parts, $\omega_\nu = |\omega_\nu|e^{i\phi_\nu}$. The matrix $\rho_A$ can be diagonalized by $\mathbb{L}^\dagger$ and $\mathbb{R}$, $\mathbb{L}^\dagger\mathbb{R} = \mathbb{I}$, such that

$$\mathbb{L}^\dagger\rho_A\mathbb{R} = \text{diag}(\omega_\nu) = \text{diag}(|\omega_\nu|)\text{diag}(e^{i\phi_\nu}).$$

The amplitude and phase parts of the reduced density matrix are defined as $|\rho_A| := \mathbb{R}\,\text{diag}(|\omega_\nu|)\mathbb{L}^\dagger$ and $e^{i\Phi} := \mathbb{R}\,\text{diag}(e^{i\phi_\nu})\mathbb{L}^\dagger$. In this notation, $\rho_A = |\rho_A|e^{i\Phi}$.

Now we give the generic definitions of the entanglement entropy and the $n$-th Rényi en-

---

this approach is not a trivial function of $L_A$. For the conformal vacuum $h = 0$, it will reduce to the usual form $S_A^{(n)}(L_A) = \frac{c}{6}(1+\frac{1}{n})\ln(L_A) + \cdots$.

tropy,

$$S_A := -\operatorname{Tr}(\rho_A \ln|\rho_A|) = -\sum_\nu \omega_\nu \ln|\omega_\nu|,$$

$$S_A^{(n)} := \frac{1}{1-n} \ln\left(\operatorname{Tr}(\rho_A|\rho_A|^{n-1})\right) = \frac{1}{1-n} \ln\left(\sum_\nu \omega_\nu|\omega_\nu|^{n-1}\right). \tag{1}$$

The above definitions give the desired properties of the entanglement measures. First, both the *generic* entanglement entropy and the *n*-th Rényi entropy have the correct definition in the Hermitian limit, i.e., $|\rho_A| = \rho_A$, $e^{i\Phi} = \mathbb{I}$. Second, the *generic* first Rényi entropy is equal to the *generic* entanglement entropy, $S_A^{(n=1)} = \lim_{n\to 1} \frac{1}{1-n} \ln \operatorname{Tr}\rho_A|\rho_A|^{n-1} = -\partial_n \operatorname{Tr}(\rho_A|\rho_A|^{n-1})\big|_{n=1} = S_A$. Third, for all the cases we studied, the eigenvalues of the reduced density matrix are real or conjugate pairs, which lead to the real outcomes. The definitions of the *generic* entanglement and Rényi entropies [Eq. (1)] are the main results in this work.

We validate the *generic* entanglement and Rényi entropies capture the correct entanglement properties from various critical non-Hermitian systems. At the critical point, this non-Hermitian system can be described by the non-unitary conformal field theory with the negative central charge. We compute both the *generic* entanglement and Rényi entropies for different critical non-Hermitian models. All results give the expected scaling as a function of the subsystem size. We also demonstrate the equivalency of our definition with the modified trace formalism in the quantum group symmetric spin chains [20] in the Appendix C. Finally, we compare the traditional and *generic* entanglement/Rényi entropy in the AKLT model with the non-Hermitian perturbation and demonstrate the validity of our proposed definition.

## 2.1 The non-Hermitian two-legged SSH model at critical points

We consider the two-legged SSH with the single-particle Hamiltonian in the momentum space as,

$$\mathcal{H}(k) = \begin{bmatrix} iu & \eta \\ \eta^* & -iu \end{bmatrix}, \quad \eta = -w - v_1 e^{-ik} - v_2 e^{ik}, \tag{2}$$

where $w$ and $v_{1(2)}$ are the intra- and inter-cell hopping terms, and $u$ is the imaginary chemical potential [Fig. 1a]. The research on the SSH model is widely extended [24–29]. Due to its simplicity and nontrivial topological properties, it has been treated as a parent model for studying many-body systems with non-Hermiticity. The many-body ground state of this model is considered as filling all negative single-particle energy modes. The phase diagram of this model is shown in Fig. 1b, containing three parity and time-reversal ($\mathcal{PT}$) preserving phases (trivial, topological I, and II) and a $\mathcal{PT}$ broken phase, which are symmetric about $v_1 = v_2$. For either $v_1 = 0$ or $v_2 = 0$, this model reduces to the non-Hermitian SSH model [30, 31], which hosts both trivial and topological phases as in the usual SSH model, with a $\mathcal{PT}$ broken phase between them.

At the phase boundary, the system is gapless and all the single-particle energies are real. There are certain momenta $k = k_{EP}$ corresponding to degenerate energies which we denote those momenta as the exceptional points (EP)s in the momentum space. In Fig. 1c, there is one $k_{EP}$ in the energy spectrum which corresponds to the blue segments in Fig. 1b. On the other hand, there are two $k_{EP}$'s in the energy spectrum [Fig. 1d] which corresponds to the orange curves in Fig. 1b. When calculating the generic entanglement/Rényi entropy, a proper limit needs to be taken to avoid divergence at $k_{EP}$ [see Supplementary information for details].

In Figs. 1e-h, the data fitting shows that the critical behaviors of the *n*-th Rényi entropy with different *n* agrees perfectly with the logarithmic scaling [1–4] for the fixed ratio $L_A/L = 1/2$, $S_A^{(n)} = \frac{c}{6}\left(1 + \frac{1}{n}\right)\ln L_A + a_n$, and also for a fixed total system size $L$, $S_A = \frac{c}{3}\ln\left[\sin\left(\frac{\pi L_A}{L}\right)\right] + b_n$,

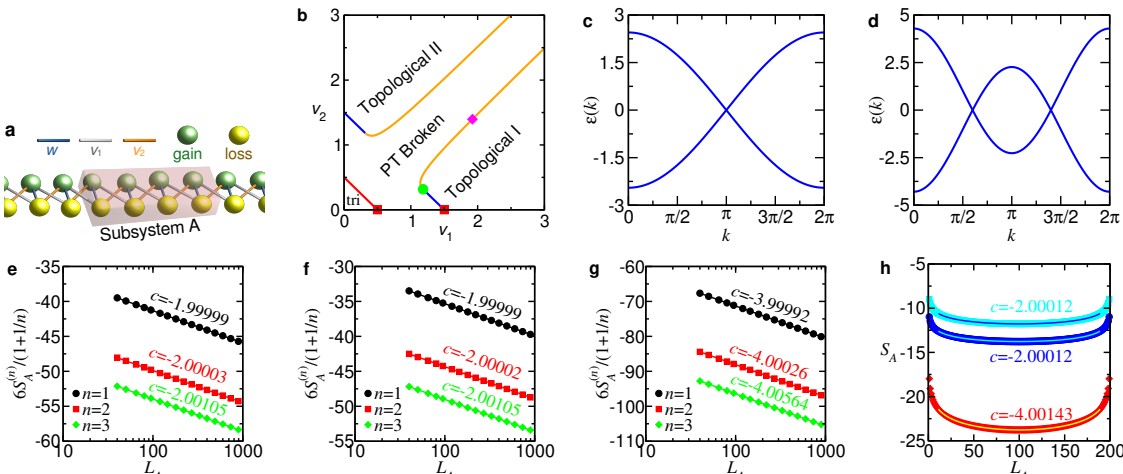

Figure 1: **The two-legged non-Hermitian SSH model. a**, The subsystem $A$ is a segment of the systems containing $L_A$ unit cells. **b**, The phase diagram of the two-legged SSH model for $(w, u) = (1, 0.5)$. **c**, and **d**, show the single-particle energy dispersion at the red point $(v_1, v_2) = (1.5, 0)$ and the magenta point, respectively. The green point is the quadratic band touching point. We discuss the entanglement/Rényi entropy at this point in the Supplementary Information. The logarithmic scaling of the $n$-th Rényi entropy $S_A^{(n)}$ with $n = 1, 2, 3$ for the subsystem sizes $L_A = L/2$ for **e**, trivial-$\mathcal{PT}$ broken transition point, **f**, topological-$\mathcal{PT}$ broken point with only one $k_{EP}$, and **g**, the transition point with two $k_{EP}$'s. **h**, the entanglement entropy for the above different critical points for a fixed total size $L = 200$. The lines are the numerical fitting curves, and the numbers shown in each figure are the fitted central charges.

where $a_n$ and $b_n$ are constants. All the fitted central charges agree well with the expected $c = -2$ in the single $k_{EP}$ cases and $c = -4$ in the two $k_{EP}$'s cases. The two-legged SSH model is a non-interacting model which is convenient to extract the true central charge from the finite-size scaling of the *generic* entanglement/Rényi entropy. In the following section, we demonstrate our proposal can also be applied to more complicated many-body non-Hermitian quantum systems.

## 2.2 The q-deformed critical XXZ spin chain with imaginary boundary terms

We consider the q-deformed XXZ spin model [20, 32] with open boundary condition [see Fig. 2**a**],

$$H = \sum_{j=1}^{L-1} \left( \sigma_j^x \sigma_{j+1}^x + \sigma_j^y \sigma_{j+1}^y + \cos\theta \, \sigma_j^z \sigma_{j+1}^z \right) + i \sin\theta \sum_{j=1}^{L-1} \left( \sigma_j^z - \sigma_{j+1}^z \right), \tag{3}$$

where $\theta \in [0, \pi/2]$ and $\sigma^\alpha$, $\alpha = x, y, z$ are the Pauli matrices. This model is non-Hermitian and critical. The Hermiticity can be restored by taking the periodic boundary condition, where the boundary imaginary terms disappear. The Hamiltonian [Eq. (3)] is the anisotropic limit of an integrable six-vertex model with complex Boltzmann weights, where the central charge can be expressed as $c = 1 - 6\frac{\theta^2/\pi^2}{1-\theta/\pi}$ which can be negative as shown in Fig. 2**b**. In the six-vertex model with complex Boltzmann weights, the phase factors cancel everywhere, except at boundaries and along lines connecting conical singularities. Due to these phase factors, the trace operation requires an modified form, which includes a factor $q^{-2\sigma_A^Z}$ [20]. On the other hand, the Hamiltonian [Eq. (3)] can also be viewed as the anisotropic limit of an integrable

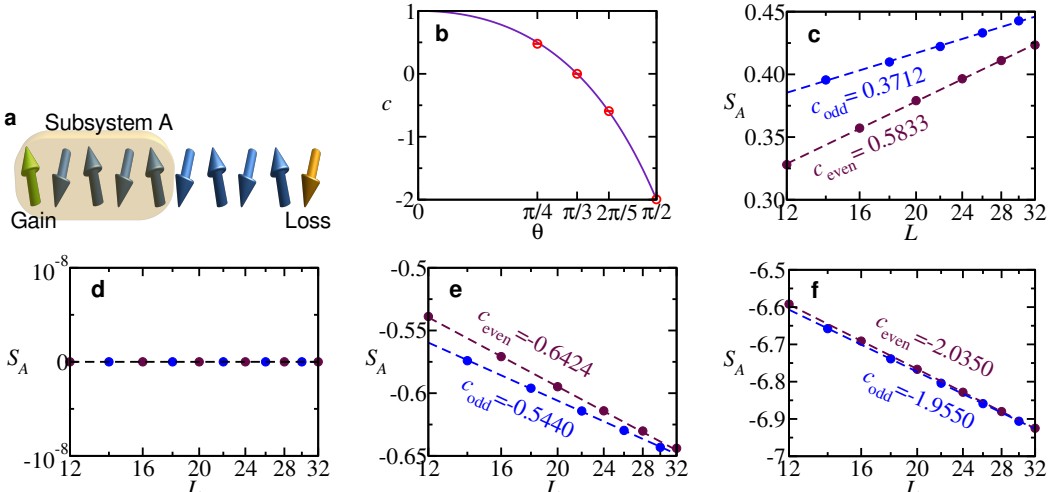

Figure 2: **The q-deformed XXZ spin-1/2 chain. a**, The non-Hermitian terms of gain/loss remain at the ends with open boundary condition. The length of subsystem A is chosen as $L_A = L/2$, such that the bipartition is far from the ends. **b**, The indigo line is the theoretical value of the central charge as a function of $\theta$, and the red circles are the average of numerical fitted central charges from $c_{odd}$ and $c_{even}$. The logrithmic scaling of the *generic* entanglement entropy for **c**, $\theta = \frac{\pi}{4}$, **d**, $\theta = \frac{\pi}{3}$, **e**, $\theta = \frac{2\pi}{5}$, and **f**, $\theta = \frac{\pi}{2} - 10^{-3}$.

six-vertex model with real Boltzmann weights together with non-trivial complex boundary condition. The corresponding central charge $c = 1$. The trace operation in this situation is just the usual trace.

For an open spin-chain, the equal bipartition is considered such that the subsystem $A$ borders with the boundary. The entanglement entropy has the scaling form $S_A \sim \frac{c}{6} \ln L$. We use the Lanczos exact diagonalization to compute the left and right ground states for the system size up to $L = 32$ [See Appendix A]. The central charges are extracted from the scaling behavior of the *generic* entanglement entropy. As shown in Figs. 2**b**-**f**, the central charge from the finite-size scaling of the *generic* entanglement entropy shows nice agreement with the analytic form. We also consider the case of fixing the total system size $L$ and varying the subsystem size $L_A$. The scaling behavior of the *generic* entanglement entropy has the form $S_A \sim \frac{c}{6} \ln\left[\sin\left(\frac{\pi L_A}{L}\right)\right]$. The corresponding central charges extracted from this case are identical to the equal bipartite case.

## 2.3 The pseudo-Hermitian AKLT model

Finally, we consider the fully gapped interacting spin chain for further checking the validity of the *generic* entanglement/Rényi entropy. The AKLT model with the pseudo-Hermitian perturbation is

$$H = \sum_{j=1}^{L}\left(\mathbf{S}_j \cdot \mathbf{S}_{j+1} + \frac{1}{3}(\mathbf{S}_j \cdot \mathbf{S}_{j+1})^2\right) + i\gamma S_{L-1}^z S_L^z S_1^z, \tag{4}$$

where $\mathbf{S}_j$ are the spin-1 operators at the $j$-th site and $\gamma \in \mathbb{R}$. The pseudo-Hermitian property of the Hamiltonian is $\eta H \eta^{-1} = H^\dagger$ with $\eta : S_i^z \to -S_i^z$. At $\gamma=0$, the exactly solvable AKLT model [33] is fallen into the symmetry-protected-topological (SPT) phase, protected by the parity $\mathcal{P}$, time-reversal $\mathcal{T}$, and $\mathbb{Z}_2 \times \mathbb{Z}_2$ symmetries [34–37]. The ground-state is described by the valence bond solid with the valence bond connecting fractionalized spin-1/2 at each site. A

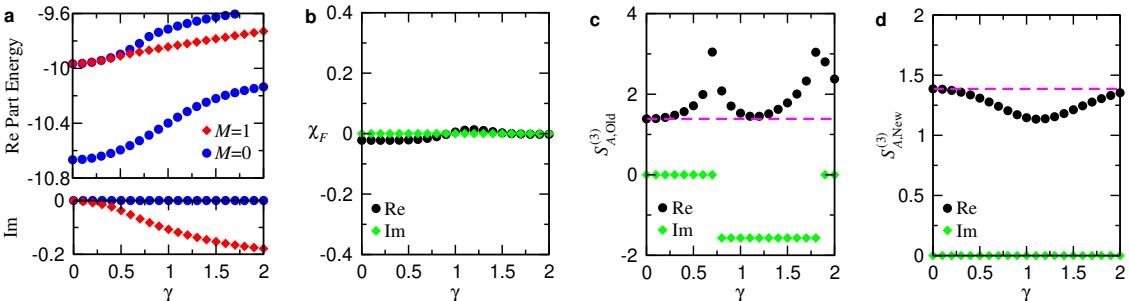

Figure 3: **The pseudo-Hermitian AKLT model.** **a**, The real and imaginary part of the lowest energy spectra for the magnetization $M = 0$ and $M = 1$ as functions of $\gamma$. **b**, The fidelity susceptibility of almost zero detects neither quantum critical point nor EP. **c**, The traditional third Rényi entropy $S_{A,\text{Old}}^{(3)}$ and **d**, the generic third Rényi entropy $S_{A,\text{New}}^{(3)}$ as a function of $\gamma$. The dashed lines are $2\ln 2$. The system size $L = 16$ and the subsystem size $L_A = 8$. The *generic* Rényi entropy is defined by Eq. (1), while the traditional Rényi entropy is $S_{A,\text{Old}}^{(n)} = \frac{1}{1-n} \ln\left(\text{Tr}\rho_A^n\right)$.

pair of spin-1/2's in the adjacent sites form the singlet state $\frac{1}{\sqrt{2}}(|\uparrow\downarrow\rangle - |\downarrow\uparrow\rangle)$, and two spin-1/2's at each site are projected into the spin-1 subspace. The magnon excitation is a triplet with a Haldane gap and separated from the singlet ground-state. Although the SO(3) symmetry is broken by the $\gamma$ term which eliminates the triplet degeneracy, the magnetization $M = \sum_j S_j^z$ is a good quantum number. However, only the $M = 0$ sector has the real eigenvalue, and the eigenvalues for $\pm M$ sectors are the complex conjugate pairs.

In Fig. 3**a**, the ground state energy is real and the energy gap between the ground state and the first excited state is finite. Since the ground state is intact due to the finite gap, one expects the entanglement/Rényi entropy is a smooth function of $\gamma$. Further examination of non-singular behavior in the parameter space can be made by considering the fidelity susceptibility $\chi_F(\gamma) = \frac{1 - \mathcal{F}(\gamma)}{L\epsilon^2}$, where the fidelity $\mathcal{F}(\gamma) = \langle\psi_0^L(\gamma)|\psi_0^R(\gamma + \epsilon)\rangle\langle\psi_0^L(\gamma + \epsilon)|\psi_0^R(\gamma)\rangle$ is the inner-product of the left/right ground states with nearby parameters $\gamma$ and $\gamma + \epsilon$. We take $\epsilon = 10^{-3}$. The fidelity susceptibility is found to be able to detect both the quantum critical point ($\chi_F = +\infty$) and the EP ($\chi_F = -\infty$) [38, 39]. As shown in Fig. 3**b**, the fidelity susceptibility of nearly zero indicates the absence of both quantum critical point and the EP in the parameter space.

We compute the entanglement/Rényi entropy from the traditional definition and the new definition [Eq. (1)] in the region $|\gamma| < 2$. Although both the traditional and *generic* $S_{A,\text{Old/New}}^{(n)}$ for $n = 1, 2$ are smooth functions of $\gamma$ [see Supplementary information for details], the traditional 3$^{\text{rd}}$ Rényi entropy has singularities as shown in Fig. 3**c**. These unnatural singularities come from $\text{Tr}\rho_A^3 = 0$ where there is an accidentally perfect cancellation of the positive and negative eigenvalues of $\rho_A$. On the other hand, no cancellation is found in the *generic* third Rényi entropy [Fig.3**d**]. From our results of various non-Hermitian systems, we expect that the *generic* Rényi entropy remains a smooth function when the ground state energy is real and the system has a finite gap, while the traditional Rényi entropy can have singularities due to the improper definition.

## 3 Conclusion and discussion

In this work, we establish a connection between quantum information science and non-Hermitian quantum systems. The *generic* entanglement and Rényi entropies capture the correct entanglement properties in non-Hermitian quantum systems. Experimentally, these non-Hermitian quantum systems can be realized as the few-body atom gain and loss [40–43] in an optical lattice. One can prepare the left and right many-body ground state and measure the *generic* entanglement/Rényi entropy by quantum state tomography [44, 45]. In principle, one can measure the true central charge for many-body non-Hermitian quantum systems at the phase transition point. Also, in Ref. [46], the Ising model with an imaginary field (Yang-Lee edge singularity [47]) can be embedded in a quantum system with an ancilla qubit. The non-unitary criticality can be considered as the post-selection of quantum measurements. One should be aware that the entanglement/Rényi entropy conditioned on post-selection can be negative [48–50]. We expect the *generic* entanglement/Rényi entropy can be related to the conditional entanglement/Rényi entropy. It is interesting to note that the *generic* Rényi entropy can be expressed as

$$S_A^{(n)} = \frac{1}{1-n} \ln\left(\text{Tr}(\rho_A^{m+1} \rho_A^{\dagger m})\right), \quad m = \frac{n-1}{2}. \tag{5}$$

From the field theory treatment with $m \in \mathbb{Z}^+$, Eq. (5) is a partition function on the $(2m + 1)$-sheeted surface with the partial time-reversal operation applied on subsystem $A$ along the $m$ copies.

Although our main focus is the non-Hermitian systems which can be described by non-unitary CFTs, our proposal can apply to many non-Hermitian quantum systems. In particular, for the quadratic band touching point in the two-legged SSH model which cannot be described by CFTs, we observe that the *generic* entanglement entropy also has a logarithmic scaling with effective central charge $c = -6$ [see Appendix B.2]. It is still desired to have the effective field theory description of understanding the universal entanglement properties of the quadratic band touching point or the Lifshitz transition in non-Hermitian systems. Furthermore, the entanglement dynamics, the holographic duality, and the robustness of the topological entanglement entropy in non-Hermitian systems are interesting subjects for future investigation. We believe that our work paves a new direction to study the entanglement properties in non-Hermitian quantum systems.

## Acknowledgements

We thank Pochung Chen, Chang-Tse Hsieh and Yi-Ping Huang for insightful discussions. We also thank the anonymous Referee for pointing out the expression in Eq. (5). PYC is supported by the Young Scholar Fellowship Program by Ministry of Science and Technology (MOST) in Taiwan. We also thank NCTS for their support.

**Author contributions** Y.-T. T. and P.-Y. C. developed the theoretical expressions. Y.-C. T. performed the large scale numerical calculation. All the authors contributed to the discussion and the preparation of the manuscript.

**Funding information** This work is supported by the MOST under grants No. 110-2636-M-007-007.

# A  Numerical Methods

Utilizing the non-Hermitian Lanczos exact diagonalization method for complex symmetric matrix [51], we compute the ground state eigenvalue and the corresponding left/right eigenvectors for the q-deformed spin-1/2 XXZ chain and the spin-1 AKLT model with non-Hermitian perturbations. For both spin models, the magnetization in the $z$-direction is a good quantum number. Moreover, the reduced density matrix commutes with the subsystem magnetization. Thus, both the Hamiltonian and the reduced density matrix can be block diagonalized by magnetization. The first excited energy in the AKLT model is computed by a few extra iterations. The dimension of the Hilbert space in the sector $M{=}0$ for the AKLT model with $L{=}16$ is $\mathcal{D}{=}5196627$, and for the q-deformed XXZ chain with $L{=}32$ is $\mathcal{D}{=}601080390$. To converge to the high accuracy of eigenvectors, the restart process with the final vector as the initial vector is usually needed. For the q-deformed XXZ chain with $\theta{=}\pi/2{-}10^{-3}$, this tiny shift prevents the numerical difficulty from the ill-conditioned point $\theta{=}\pi/2$. The condition number for $L{=}32$ with $\theta{=}\pi/2{-}10^{-3}$ is about $7{\times}10^4$. In this case, carefully choosing the random initial vector is important and tedious for trying. After obtaining the accurate eigenvectors, the imaginary part of the generic entanglement entropy is about $-6{\times}10^{-7}$ for $L{=}32$ in the q-deformed XXZ chain with $\theta{=}\pi/2{-}10^{-3}$, which is the worst-case compared with the other cases. Therefore, we confidently consider all the generic entropies are real, with ignorable numerical round-off error of imaginary part. Parallelized numerical computations were performed on the High-Performance Computing (HPC)-cluster with Intel i9-10900K CPU and 64GB memory.

# B  Details of the two-legged SSH model

The non-Hermitian two-legged SSH model with $\mathcal{PT}$ symmetry is

$$H = -w\sum_{j=1}^{L}(c_{j\uparrow}^{\dagger}c_{j\downarrow} + \text{H.c.}) + iu\sum_{j=1}^{L}(n_{j\uparrow} - n_{j\downarrow}) - v_1\sum_{j=1}^{L}(c_{j\uparrow}^{\dagger}c_{j+1\downarrow} + \text{H.c.}) - v_2\sum_{j=1}^{L}(c_{j\downarrow}^{\dagger}c_{j+1\uparrow} + \text{H.c.}),$$

(6)

where $c_{j\uparrow}$ and $c_{j\downarrow}$ are the annihilation operators at the $j$th site for the leg with gain and loss, respectively. $u$ is the parameter for non-Hermiticity, and $n_{j\sigma} = c_{j\sigma}^{\dagger}c_{j\sigma}$ is the number operator. Periodic boundary conditions are assumed, i.e., $c_{L+1\sigma} := c_{1\sigma}$. By employing Fourier transform, $\tilde{c}_{k\sigma} = \frac{1}{\sqrt{L}}\sum_{j=1}^{L}e^{ikj}c_{j\sigma}$, where $k = \frac{2\pi m}{L}$ and $m = 0,\dots,L-1$, the single-particle Hamiltonian $\mathcal{H}(k)$ is obtained

$$H = \sum_{k}\begin{bmatrix}\tilde{c}_{k\uparrow}^{\dagger} & \tilde{c}_{k\downarrow}^{\dagger}\end{bmatrix}\mathcal{H}(k)\begin{bmatrix}\tilde{c}_{k\uparrow}\\\tilde{c}_{k\downarrow}\end{bmatrix},$$

(7)

where the single-particle Hamiltonian $\mathcal{H}(k)$ is shown in Eq. (2).

The single-particle eigenenergies are $\varepsilon_{\pm}(k) = \pm\sqrt{\Delta(k)}$, where

$$\Delta(k) = v_1^2 + v_2^2 + w^2 + 2w(v_1 + v_2)\cos k + 2v_1 v_2\cos 2k - u^2,$$

and the biorthogonal left/right eigenvectors for $\Delta(k) > 0$ ($\mathcal{PT}$ preserving) are

$$|L_{\pm}(k)\rangle = \frac{1}{\sqrt{2\Delta(k)}}\begin{bmatrix}iu \mp \sqrt{\Delta(k)}\\e^{ik}v_1 + e^{-ik}v_2 + w\end{bmatrix},$$

$$|R_{\pm}(k)\rangle = \frac{1}{\sqrt{2}(\pm iu + \sqrt{\Delta(k)})}\begin{bmatrix}-iu \mp \sqrt{\Delta(k)}\\e^{ik}v_1 + e^{-ik}v_2 + w\end{bmatrix}.$$

The half-filled ground state $|\psi_0^{L/R}\rangle$ is constructed by filling all negative energy modes.

Figs. 1**c** and **d** in the main text show the single-particle energy dispersion at various points at the phase boundaries. At the boundary between the trivial phase and the $\mathcal{PT}$ broken phase, there is a single $k_{EP} = \pi$. The behavior is the same at the part of the boundary between the $\mathcal{PT}$ broken phase and the topological phase shown in the blue line segments in Fig. 1**b** in the main text. For the part shown in the orange curve, two $k_{EP}$'s appear symmetrically about $k = \pi$ [see Fig. 1**d** in the main text]. Approaching the green point along the orange curve, these $k_{EP}$'s approach $k = \pi$ and become a single $k_{EP} = \pi$ with the quadratic band touching [Fig. 4**a**].

In calculations, careful selection of sizes and parameters is important, so that the properties of conformal field theory can be retained. Firstly, there should be a $k$-mode very close to each $k_{EP}$. Otherwise, the effect of the EPs would disappear in the finite system. Secondly, since $k = k_{EP}$ causes a singularity $(1/\sqrt{\Delta(k)} = \infty)$ in the calculation of the entanglement entropy, we need a tiny shift from $k_{EP}$, but keeping the scale invariance of the system as the system size $L$ changes.

In the case of a single $k_{EP} = \pi$, we choose the momenta $k = \frac{2\pi m + \delta}{L}$, $m = 0, \ldots, L-1$, $\delta \ll 1$, such that $k = k_{EP} + \delta/L$ for $m = L/2$. Note that we put $\delta$ in this way so that no additional length scale is introduced. In the case of a pair of $k_{EP} \neq \pi$, the parameters are chosen such that $k_{EP}/\pi = 1 \pm p/q$ is a rational number, and the system size $L$ is chosen as a multiple of $2q$. With this choice, we have $k = k_{EP} + \delta/L$ for some $m$.

Followed by the above selection rules, we choose the parameters $(v_1, v_2) = (0.5, 0)$ and $(1.5, 0)$ for the trivial–to–$\mathcal{PT}$-broken and $\mathcal{PT}$-broken–to–topological cases, respectively (the red dots in Fig. 1**b** in the main text), and $(v_1, v_2) \approx (1.9220798186197803, 1.3970417517659157)$, corresponding to $k_{EP}/\pi = 1\pm2/5$, (purple dot in Fig. 1**b** in the main text) for the case that a pair of $k_{EP} \neq \pi$. In all cases, $\delta = 10^{-5}$ is used.

To calculate the entanglement entropy, we need the reduced density matrix $\rho_A$ for a subsystem $A$, which can be obtained by the overlap matrix $M^A$ [22]. The matrix element of $M^A$ is calculated by

$$M_{\alpha\beta}^A = \sum_{i \in A} \mathbb{L}_{\alpha,i}^\dagger \mathbb{R}_{\beta,i}, \quad \alpha, \beta \in \text{occupied modes}, \tag{8}$$

where $\mathbb{L}_{\alpha,i}$ and $\mathbb{R}_{\alpha,i}$ are the left and right spatial wavefunction of the occupied mode $\alpha = (k, -)$, respectively. We have

$$\rho_A = \bigotimes_\nu \left( \lambda_\nu |L_\nu^A\rangle\langle R_\nu^A| + (1 - \lambda_\nu)|0\rangle\langle 0| \right), \tag{9}$$

where $\left|L_\nu^A\right\rangle$ and $\left|R_\nu^A\right\rangle$ are the left and right biorthogonal eigenvectors of $M^A$ with the eigenvalues $\lambda_\nu$, respectively. $\left\langle L_\nu^A | R_\mu^A \right\rangle = \delta_{\nu\mu}$, and $\delta_{\nu\mu}$ is a Kronecker delta function. The entanglement entropy are then calculated after the eigenvalues $\lambda_\nu$ are obtained. For the Hermitian free-fermion systems, see Ref. [52].

We also calculate the entanglement entropy by using the correlation matrix method [22, 53]. The matrix elements $C_{ij}$ of the correlation matrix $C$ is $C_{ij} = \langle\psi_0^L|c_i^\dagger c_j|\psi_0^R\rangle$, where $i, j \in A$. By the formulas modified from the Hermitian case [25, 53–55], we have:

$$S_A = -\sum_\nu (\lambda_\nu \ln|\lambda_\nu| + (1 - \lambda_\nu)\ln|1 - \lambda_\nu|), \tag{10}$$

$$S_A^{(n)} = \frac{1}{1-n}\sum_\nu \ln\left(\lambda_\nu|\lambda_\nu|^{n-1} + (1 - \lambda_\nu)|1 - \lambda_\nu|^{n-1}\right). \tag{11}$$

Where $\lambda_\nu$ is the $\nu$-th eigenvalue of the correlation matrix $C$ or the overlap matrix $M^A$.

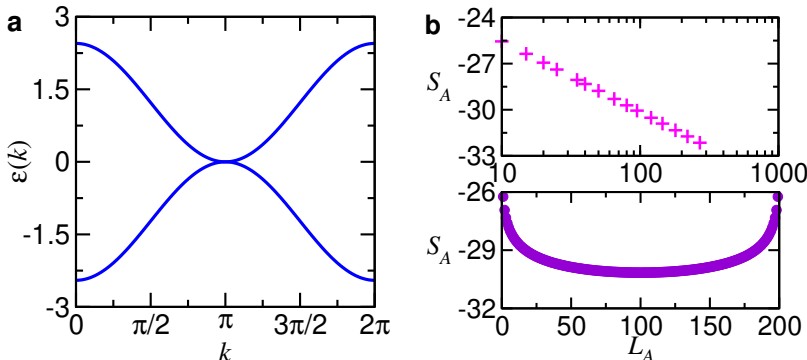

Figure 4: **a**, The quadratic band touching takes place at the green point in the phase diagram in Fig. 1**b** in the main text. At this point, we do not expect conformal symmetry. **b**, The logarithmic scaling of the entanglement entropy at the quadratic band touching point.

## B.1 Dependence on the momentum shift

As we discussed in the previous section, in the two-legged non-Hermitian SSH model, to avoid the singularity, we need to introduce a momentum shift $\delta$ for computing the generic entanglement/Rényi entropy. Here we summarize the dependence of $\delta$ in $S_A^{(n)}$. If we fix $L_A/L = 1/2$, and change $\delta$, fitting suggests the following behavior (including $S_A = S_A^{(n=1)}$):

1. For $c = -2$ cases, $S_A^{(n)} = \frac{-2(n+1)}{6n} \ln L_A + \ln \delta + a_n$.

2. For $c = -4$ cases, $S_A^{(n)} = \frac{-4(n+1)}{6n} \ln L_A + 2\ln \delta + b_n$.

3. At the quadratic band touching point, $S_A^{(n)} = -2\ln L_A + 2\ln \delta + \tilde{c}$.

Here $a_n$, $b_n$, and $\tilde{c}$ are constants. We let $\delta = e^{-d}$ and fit $S_A^{(n)}$ versus $d$ for each of $n = 1, 2, 3$, $L_A = 50, 100, 150$, and $(v_1, v_2)$ at each of the three points calculated in the main text (with $d = 20, 30, 40, 50$) and the quadratic band touching point (with $d = 10, 15, 20, 25$ due to numerical limitation). In each case, the coefficient of $\ln \delta$ equals the above coefficient up to at least 9 decimal digits.

## B.2 The behavior near the quadratic band touching point

In the phase diagram of the two-legged SSH model [Fig. 1(b) in the main text], there is a quadratic band touching point (green dot) separating the PT-broken–to–topological phase boundary with $c = -2$ [blue line segment in Fig. 1(b) in the main text] and that with $c = -4$ [orange curve in Fig. 1 in the main text]. Its coordinate is

$$(v_1^Q, v_2^Q) \approx (1.1830127018922194, 0.3169872981077807). \tag{12}$$

We summarize the behavior as we go along the blue line, pass through this quadratic band touching point, and then go along the orange curve. In each case, we calculate $S_A^{(n)}, n = 1, 2, 3$ for $L_A$ up to the order of $10^3$ with $\delta = 0.00002$.

1. Along the $c = -2$ phase boundary [blue line] with $v_2 - v_2^Q \approx -10^{-4}$:

   $S_A^{(n)}$ shows the expected asymptotic behavior of $c = -2$ for large $L_A$. For small $L_A$, it behaves like item 2. below.

2. Along the $c = -2$ phase boundary [blue line] with $v_2 - v_2^Q \approx -10^{-8}$:

   $S_A^{(n)}$ is dominated by a single pair of eigenvalues, $1/2 \pm \alpha i$, of both the overlap and the correlation matrix. This means that $S_A^{(n)} \approx -2 \ln \left| \frac{1}{2} + \alpha i \right|$ is almost independent of $n$. $S_A^{(n)}$ shows log scaling with $L_A$ with coefficient $-1$. The behavior of $c = -2$ is expected to be retained at extremely large $L_A$, but is beyond numerical calculation.

3. Along the $c = -2$ phase boundary [blue line] with $v_2 - v_2^Q \approx -10^{-15}$:

   $S_A^{(n)}$ behaves like item 2. for large $L_A$. For small $L_A$, it behaves like item 4. below.

4. At the quadratic band touching point, $(v_1^Q, v_2^Q)$:

   Both the overlap and the correlation matrix shows only (up to numerical error) a single pair of eigenvalues, $1/2 \pm \alpha i$, other than 0 and 1. This means that $S_A^{(n)} = -2 \ln \left| \frac{1}{2} + \alpha i \right|$ is independent of $n$. $S_A^{(n)}$ shows log scaling with $L_A$ with coefficient exactly $-2$ (up to numerical error). In the main text, we refer this coefficient to the effective central charge $c = -6$.

5. Along the $c = -4$ phase boundary [orange curve] with $v_2 - v_2^Q \approx +10^{-4}$

   $S_A^{(n)}$ shows the expected asymptotic behavior of $c = -4$ for large $L_A$. Unlike the previous cases, we cannot explore the behavior of very small $L_A$. Since $k_{\text{EP}}/\pi \approx 1 \pm 10^{-2}$, $L_A$ must be multiples of at least several hundreds. In such a scale, the behavior of $S_A^{(1)}$ is already quite close to the expected asymptotic behavior of $c = -4$, but $S_A^{(n)}$ for larger $n$ scales more like item 2. above.

## C    The equivalency of the generic entanglement entropy and the quantum group entanglement entropy

Let us consider a critical quantum group symmetric XXZ spin chain with the following Hamiltonian,

$$H = -\sum_i e_i, \quad e_i = -\frac{1}{2} [\sigma_i^x \sigma_{i+1}^x + \sigma_i^y \sigma_{i+1}^y + \frac{q + q^{-1}}{2} (\sigma_i^z \sigma_{i+1}^z - 1) + h_i], \qquad (13)$$

where $q \in \mathbb{C}$, $|q| = 1$, and $h_i = (q - q^{-1})(\sigma_i^z - \sigma_i^z)/2$. Suppose we consider two sites and restrict the phase of $q$, $\text{Arg}(q) \in [0, \pi/2]$, the Hamiltonian $H$ is not Hermitian but the spectrum is real. The corresponding right ground state and excited state are

$$|\psi_0^R\rangle = \frac{1}{\sqrt{q + q^{-1}}} (q^{-1/2} |\uparrow\downarrow\rangle - q^{1/2} |\downarrow\uparrow\rangle), \quad |\psi_1^R\rangle = \frac{1}{\sqrt{q + q^{-1}}} (q^{1/2} |\uparrow\downarrow\rangle + q^{-1/2} |\downarrow\uparrow\rangle), \quad (14)$$

with the eigenenergies $E_0 = -(q + q^{-1})$ and $E_1 = 0$, respectively. The left eigenvectors are obtained from changing $q \to q^{-1}$ in Eq. (14). They satisfy the biorthonormal condition, $\langle i^L | j^R \rangle = \delta_{ij}$ for $i, j = 0, 1$. The density matrix can be constructed from the left and right ground states

$$\rho = |\psi_0^R\rangle\langle\psi_0^L| = \frac{1}{q + q^{-1}} \begin{bmatrix} 0 & 0 & 0 & 0 \\ 0 & q^{-1} & -1 & 0 \\ 0 & -1 & q & 0 \\ 0 & 0 & 0 & 0 \end{bmatrix}. \qquad (15)$$

In Ref. [20], the modified trace is introduced for the consideration in quantum group symmetric spin chains. The reduced density matrix from the modified trace formula is

$$\tilde{\rho}_A = \text{Tr}_{\bar{A}}(q^{-2\sigma_{\bar{A}}^z}\rho) = \frac{1}{q+q^{-1}}\begin{bmatrix} 1 & 0 \\ 0 & 1 \end{bmatrix}. \tag{16}$$

The modified trace gives the correct normalization $\text{Tr}_A(q^{-2\sigma_A^z}\tilde{\rho}_A) = 1$. The entanglement entropy computed from the modified trace is

$$\tilde{S}_A = -\text{Tr}\left((q^{-2\sigma_A^z}\tilde{\rho}_A)\ln\tilde{\rho}_A\right) = \ln(q+q^{-1}). \tag{17}$$

On the other hand, one can construct the reduced density matrix in the ordinary way

$$\rho_A = \text{Tr}_{\bar{A}}(\rho) = \frac{1}{q+q^{-1}}\begin{bmatrix} q & 0 \\ 0 & q^{-1} \end{bmatrix}. \tag{18}$$

As one expects, it also satisfies the normalization $\text{Tr}_{\bar{A}}\rho_A = 1$. Now we can compute the generic entanglement entropy defined in the main text,

$$\begin{aligned} S_A = -\text{Tr}(\rho_A \ln|\rho_A|) &= -\left(\frac{q}{q+q^{-1}}\ln\left|\frac{q}{q+q^{-1}}\right| + \frac{q}{q+q^{-1}}\ln\left|\frac{q^{-1}}{q+q^{-1}}\right|\right) \\ &= \ln|q+q^{-1}| = \ln(q+q^{-1}). \end{aligned} \tag{19}$$

Here $|q| = 1$ and $\text{Arg}(q) \in [0,\pi/2]$ ensures $(q+q^{-1}) = |q+q^{-1}|$. Hence the generic entanglement entropy is identical to the entanglement entropy computed from the modified trace in the quantum group symmetric spin models.

## D  Traditional and generic entanglement and Rényi entropies in the AKLT model with non-Hermitian perturbation

We compute the generic entropy $S_{A,\text{New}}^{(1,2)}$ and the traditional entropy $S_{A,\text{Old}}^{(1,2)}$ in the presence of the non-Hermitian breaking term $\gamma \neq 0$. Both traditional and generic entanglement entropies are real and smooth functions of $\gamma$ as shown in Figs. 5(a) and (b). Similarly, the traditional and the generic 2nd Rényi entropies are real and smooth functions of $\gamma$ as shown in Figs. 5(c) and (d). However, as we discussed in the main text, the traditional 3rd Rényi entropy has singularities due to its improper definition, while the generic 3rd Rényi entropy remains a smooth function of $\gamma$.

The singularities in the traditional 3rd Rényi entropy come from $\text{Tr}\rho_A^3 = 0$. We expect these singularities do not happen for the generic $n$-th Rényi entropy. In Fig. 6, we systematically compute $W = \text{Tr}\rho_A|\rho_A|^{n-1}$ for $n = 2,\cdots,10$ in the parameter region $|\gamma| < 2$ and do not observe any singularity. Moreover, no significant change is found by doubling the system size from $L=8$ to $L=16$. we believe larger system sizes $L > 16$ do not change the validity of the generic entanglement entropy and Rényi entropies.

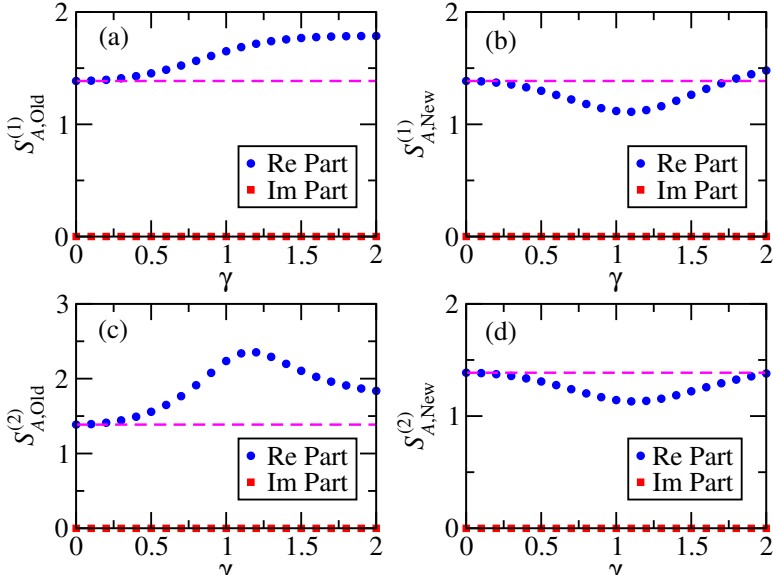

Figure 5: (a) The traditional entanglement entropy $S^{(1)}_{A,\text{Old}}$ and (b) the generic entanglement entropy $S^{(1)}_{A,\text{New}}$ as the functions of $\gamma$. (c) The traditional second Rényi entropy $S^{(2)}_{A,\text{Old}}$ and (d) the generic second Rényi entropy $S^{(2)}_{A,\text{New}}$ as the functions of $\gamma$. The dashed lines are $2\ln 2$. We choose the total system size $L = 16$ and compute the $S^{(n)}_{A,\text{Old/New}}$, $n = 1, 2$ with the subsystem size $L_A = 8$.

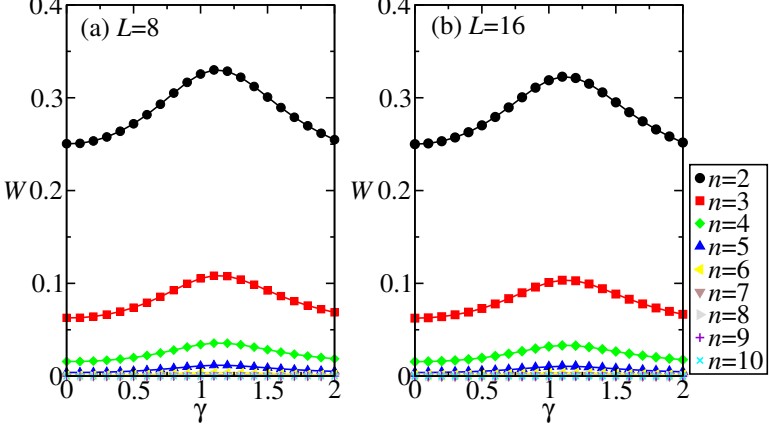

Figure 6: No singularity is found in the trace, $W = \text{Tr}\rho_A |\rho_A|^{n-1}$, for $n = 2, \cdots, 10$. (a) $L=8$ and $L_A=4$, and (b) $L=16$ and $L_A=8$. No significant change is found by doubling the system size from $L=8$ to $L=16$.

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
