# Peer review of "Rényi entropies and negative central charges in non-Hermitian quantum systems"

_SciPost Physics, doi:SciPost Phys. 12, 194 (2022)_

## Round 4 · Referee Report · Anonymous · 2022-3-29

Strengths
*The paper presents an interesting new definition of standard entanglement measures for non-hermitian systems. The definitions are a priori general, only dependent on the eigenvalues of the reduced density matrix.
*The authors back up their proposal with numerical results for several non-hermitian models.
Weaknesses
*I find that the authors slightly misrepresent their contribution to the subject (namely, defining a consistent entanglement measure for non-Hermitian quantum systems).
*Both in their abstract and in other places in the paper they give the impression that they are the first team to address the problem and also that previous proposals are somehow wrong (at least they repeatedly refer to their proposal as the "correct" or "proper" definition).
*I think there is scope to present a more nuanced discussion of what has been done before.
Report
In this paper the authors present a general proposal for the description of the entanglement entropies of non-Hermitian quantum systems. Their proposal is distinct from some of the proposals that have come before and so provides a new viewpoint on the generic question of how to define consistent measures of entanglement in non-Hermitian quantum systems.
In my view the paper is of good quality and provides a good numerical analysis of the formulae that are proposed, by considering several interesting models. Overall I think the work should be published in SciPost.
However, I do think that there should be a slightly more balanced discussion of contributions by other groups, as I feel that, perhaps inadvertently, those proposals are presented in an unfairly negative light.
I also think that the authors do not really address some of the issues their proposal has, which are still rather fundamental questions about what an entanglement entropy should be. Let me summarise my main points below:
1) In the introduction the authors write "The negative entanglement entropy seems problematic because the reduced density matrix is positive semi-definite which cannot give rise to a negative value of the entanglement entropy. "
I think this sentence is a bit unclear. The point is that by defining the reduced density matrix as the authors do in the first line of Section 2, that is in terms of right and left eigenvectors which can be distinct for non-hermitian systems, automatically such density matrix is not positive-definite in general (as the authors say, it can even have complex eigenvalues). So, I think the correct statement would be that "Defining a reduced density matrix which involves both right and left eigenvectors one naturally finds an entropy which is no longer guaranteed to be positive (or even real!)"
2) From the sentence quoted in 1) it sounds as if the authors recognise that getting a negative entropy is problematic. Then they write that
"To reconcile this issue, Refs. [17–19] suggest that the entanglement entropy is still positive and the true central charge is replaced by an effective central charge ceff which is positive [17]." and this is correct. In these papers, a different reduced density matrix is chosen so that the entropy is again guaranteed to be positive.
3) The next sentence in the introduction says "However, a proper redefinition of the entanglement measures should be considered in the non-Hermitian systems, and the correct properties such as the negative central charges can be obtained."
I find this sentence a bit inconsistence with what came earlier. First of all, in point 1) the authors seemed to recognise that there is something a bit unphysical about having negative entropies. In this sentence however they now say that such negative entropies are the "proper" entropies obtained from a "proper definition", which also suggests the definitions of previous works are all "not proper".
In the context of the entanglement of non-hermitian quantum systems, I think that it is not clear, even after the present work, what a "proper definition" of entanglement should be and in fact the existing proposals in the literature precisely differ in what they consider to be a "proper definition". Whereas the works [17]-[19] aim for a measure that is positive-definite and that still encodes universal information about the CFT, the authors of [21] instead argue that the "proper definition" of the entanglement entropy is the one following from the reduced density matrix in the first sentence of Section 2 of this paper. As a consequence their entropies are often negative.
I would argue that having negative entropies poses at least a question of interpretation, given the usual probabilistic meaning of entropy, so it would be worth mentioning this.
Requested changes
Following my report above I recommend the following:
1) In the discussion of previous contributions I suggest the authors mention that previous proposals were aiming for measures of entanglement with different properties from what the authors prioritise in this paper. While the works [17-19] defined a measure that is guaranteed to be positive-definite for all non-unitary CFTs, the authors of [21] proposed a measure that could be negative but which was based on the "true" reduced density matrix of the non-hermitian system. The current proposal is an addition to this body of work. It is a new measure which guarantees scaling with the true central charge, even when it is negative.
2) I think it should be mentioned somewhere that, while this definition of entanglement entropy, leads to quantities that can be measured in numerical experiments, so it is numerically useful, it still raises the usual issue of interpretation. What is the physical meaning of a negative entropy?
3) The abstract of the paper strongly suggests that this paper is the first attempt at dealing with non-hermitian systems. I suggest that the authors add a sentence in the abstract right after "Here, we propose a natural extension of entanglement and Rényi entropies to non-Hermitian quantum systems." which says something like "There have been other proposals for the computation of these quantities, which are distinct from what is proposed in the current paper."
4) In the first paragraph of page 3 the authors write "all the proposed entanglement measures in non-Hermitian systems are restricted to their specific models". I agree that what the authors propose in this paper is more general than previous treatments, however those treatments were also quite generic. I would replace "specific models" by "non-hermitian conformal field theories" which is what [17-19] and [21] really looked at.
5) One additional comment is that even if the proposal of [17] might look quite different from the authors do here, it is actually easy to see from [17] how one could get an entropy that scales with c instead of ceff. You would just need to set Delta=0 in equation (11) which means choosing not the field with smallest dimension of the CFT by the identity field. That would immediately produce scaling with c.
6) Finally, a questions about the main equations in the paper (1). I am actually a bit confused about these definitions, so maybe the authors can clarify this. They say that the eigenvalues w can be complex in general and so can be written in the Gauss form as modulus times a phase. Given that, it seems to me that the definitions (1) can produce complex results. Isn't that the case, since there is always a factor w which does not appear with a modulus? And is this a desired feature of the proposal? Or is there some assumption that the eigenvalues appear in complex conjugated pairs and so their imaginary parts will cancel in the sum? I am asking because the numerics all seem to produce negative entropies but never complex ones.
7) One additional comment that the authors might want to consider is that complex eigenvalues do not only occur for open quantum systems as a consequence of gain and loss (which seem to be most of the examples considered here). One can have complex eigenvalues even in closed systems. A well-known example is the complex periodic Ising chain that was studied by von Gehlen. https://iopscience.iop.org/article/10.1088/0305-4470/24/22/021

---

## Round 4 · Referee Report · Anonymous · 2022-4-19

Report
This paper proposes the definition of a family of entanglement measures for Hermitian and non-Hermitian systems. Their definition is a variant of the entanglement measures found in the literature. The authors claim that, in 1+1d critical systems, the entanglement measures follows the logarithmic scaling expected from conformal field theory (CFT), with a multiplicative constant given by the central charge. In support of this claim, they report some numerical results for three examples of 1+1d critical models, on the entanglement measures of a single interval. They do not give analytical arguments for their claim.
The numerical results reported in this manuscript are convincing. They show that the one-interval entanglement entropies defined by the author do follow the logarithmic scaling expected from CFT in the three examples they have chosen to study. For this reason, I think it is worth publishing this manuscript as a SciPost Physics article.
However, some inaccuracy in the short review of literature given in the introduction should be corrected. Indeed, refs [16-20] claim that the ordinary Renyi entropies of non-Hermitian models are governed by the "effective central charge" c-24h (where h is the lowest conformal weight of the model), whereas ref. [21] argues that this result relies on an incorrect determination of the corresponding partition function, and that, for example, the entanglement entropy scales as (c-12h)/3 log(L_A).
Moreover, when dealing with a quantum model defined by a 1d Hamiltonian, the notion of a "true central charge" (as referred to below eq. 3) is undefined. Indeed, fundamentally, the central charge is a feature of the stress-energy tensor, which specifies the response of the action to a local change of metrics. Hence, the central charge of a given model is well-defined only if the action (or the lattice Boltzmann weights) is given for a family of surfaces with varying metrics. In particular, for the XXZ model of eq. (3), by introducing "gauge" phase factors, one can construct distinct six-vertex models, with the same Hamiltonian limit given by eq. (3), but corresponding to different CFTs in the scaling limit. The definition of the modified trace eq. (15), taken from ref. [20] corresponds to the "gauge" associated to the CFT with central charge given below eq. (3), whereas the usual trace corresponds to the CFT with central charge c=1.
Finally, it is interesting to remark that the "generic" Renyi entropy defined by the authors reads
$1/(1-n) \log Tr[ (\rho_A^\dagger)^{(n-1)/2} (\rho_A)^{(n+1)/2} ]$
and thus, if n is an odd integer, it is given by a partition function on an n-sheeted surface (the same as for the usual Renyi entropy), with time-reversal applied along the subsystem A on (n-1)/2 copies. The numerical results reported in the manuscript show that, on the specific examples under study, this time-reversal has no effect on the scaling of the Renyi entropies.
Requested changes
1. Change the introduction to reflect correctly the debate in the literature, about entanglement entropies of non-Hermitian systems.
2. Correct the discussion of central charges in section 2.2

---

## Round 5 · Referee Report · Anonymous (Referee 1) · 2022-5-6

Strengths

Same as in my original report.

Weaknesses

These have now been addressed by introducing all changes I had requested in my original report.

Report

As I wrote in my original report, I think that the paper deserves publication in SciPost as it makes a novel and positive contribution to the development of entanglement measures that are appropriate for non-unitary systems.

The paper is well written and contains many relevant examples.

My main criticisms related to how the authors discussed their work in relation to previous contributions in the area. My general feeling was that some of their writing seemed unduly negative about previous proposals and that those were not always properly acknowledged.

I am happy the authors have considered my comments and acted on them. Consequently, I am happy with the paper in its current form and recommend publication without further changes.

---

## Round 5 · Referee Report · Anonymous (Referee 2) · 2022-5-11

Report

In this resubmission, the Introduction now correctly describes the literature on entanglement measures for non-Hermitian systems. However, the discussion of the central charge in Section 2.2 still suffers from the confusion I pointed out in my previous report -- defining a 1+1d model only by its quantum Hamiltonian does not determine the central charge, because this only gives $L_0+\bar{L}_0$, and not the full Virasoro algebra. Hence, the statement that "the Hamiltonian is the usual XXZ chain with central charge c = 1" makes no sense. Moreover, the central charge is a local bulk quantity, and hence it does not depend on boundary conditions.

To solve this confusion, I can suggest to use the following statements :

  • The Hamiltonian (3) [or its periodic variant] is the anisotropic limit of an integrable six-vertex (6V) model with complex Boltzmann weights, with central charge $c=1-\frac{6\theta^2}{\pi(\pi-\theta)}$.

  • In this complex-weight 6V model, the phase factors cancel everywhere, except at boundaries and along lines connecting conical singularities.

  • The trace operation consistent with the complex-weight 6V model is the modified trace, which includes a factor $q^{-2\sigma_A^z}$.

  • Alternatively, the Hamiltonian (3) can be viewed as the anisotropic limit of a 6V model with real Boltzmann weights, with central charge c=1, and non-trivial complex boundary conditions. The corresponding trace operation is the usual trace.

Requested changes

Correct the discussion on the central charge in Section 2.2.

---

## Round 5 · Author Response

We thank the two referees for their careful reading of the manuscript and for their comments.

Reply to Referee A

Regarding the comments 1-3, we thank the referee for pointing out our inconsistency of discussing the negative entanglement entropy. We rewrite the introduction part and point out the possibility to have negative entanglement entropy is from the new definition of the density matrix involving the left and right eigenvectors.  We also emphasize that Refs. [20-22] are the first few papers to obtained the  negative entanglement/Renyi entropy and our new proposed entanglement measures are in complementary to these existing approaches.

Reply to the requested changes

1) We mention the previous proposal Refs. [20-22] regarding to their methods the results in the revised manuscript.

2) The interpretation of the “negative”  entropy is already discussed in the conclusion and discussion. We point out the non-Hermiticity can be introduced by post-selection and the post-selected entropy can be negative.

3) We thank the referee for the suggestion. We add the sentence in the abstract that the referee suggested.

4) We thank the referee for the suggestion.  We follow the referee’s suggestion.

5) We add the footnote to clarify this point.

6) Yes, the eigenvalues of the reduced density matrix are real or conjugate pairs, which lead to the real outcomes. We add this sentence in the revised manuscript.

7) We thank the referee for pointing out this reference. We will investigate the complex periodic Ising chain in detail for future paper.

Reply to Referee B

We thank the Referee’s suggestion. We correct our inaccuracy of the review and revised the manuscript accordingly. We thank the referee for pointing out the expression of our generic Renyi entropy can have the geometric interpretation in the n-sheeted Riemann surface with partial-time reversal transformation.

Reply to the requested changes:

  1. We rewrite the introduction part according to the Referee’s suggestion.
  2. We add detailed discussion about the central charge in Section 2.2.

---

## Round 5 · List of Changes

1. We correct our misinterpretation of the literature reviews in the abstract and introduction parts. We extend the introduction part regarding to the existing proposals (Refs.~[17-21]).

  2. In section 2, we add one sentence discussing why the outcomes of the generic entanglement/Renyi entropy are real.

  3. We add detail discussion regarding the the section charge in Sec. 2.2 and add one reference. V. Pasquier and H. Saleur, Common structures between finite systems and confor- mal field theories through quantum groups, Nuclear Physics B 330(2), 523 (1990), doi:https://doi.org/10.1016/0550-3213(90)90122-T. 

  4. We add one new expression of the generic Renyi entropy (Eq. (5)) as pointing out by the second Referee.

---

## Round 6 · Referee Report · Anonymous · 2022-5-20

Report
The authors have implemented the requested changes. I recommend this manuscript for publication.

---

## Round 6 · Author Response

List of changes
In Sec. 2.2, we add the discussion regarding to the XXZ model and the complex-weight six-vertex model which the central charge c=1-6 \frac{\theta^2/\pi^2}{1- \theta / \pi}.

---

## Round 6 · List of Changes

In Sec. 2.2, we add the discussion regarding to the XXZ model and the complex-weight six-vertex model which the central charge c=1-6 \frac{\theta^2/\pi^2}{1- \theta / \pi}.

---

## Editorial Decision

published